# The Role of Immunotherapy in the Treatment of Adrenocortical Carcinoma

**DOI:** 10.3390/biomedicines9020098

**Published:** 2021-01-20

**Authors:** Izabela Karwacka, Łukasz Obołończyk, Sonia Kaniuka-Jakubowska, Krzysztof Sworczak

**Affiliations:** Department of Endocrinology and Internal Medicine, Medical University of Gdańsk, 80-952 Gdańsk, Poland; przepona@wp.pl (Ł.O.); sonia.kaniuka@gumed.edu.pl (S.K.-J.); ksworczak@gumed.edu.pl (K.S.)

**Keywords:** adrenocortical cancer, immunotherapy, immune checkpoint inhibitors

## Abstract

Adrenocortical carcinoma (ACC) is a rare epithelial neoplasm, with a high tendency for local invasion and distant metastases, with limited treatment options. Surgical treatment is the method of choice. For decades, the mainstay of pharmacological treatment has been the adrenolytic drug mitotane, in combination with chemotherapy. Immunotherapy is the latest revolution in cancer therapy, however preliminary data with single immune checkpoint inhibitors showed a modest activity in ACC patients. The anti-neoplastic activity of immune checkpoint inhibitors such as anti-cytotoxic-T-lymphocyte-associated-antigen 4 (anti-CTLA-4), anti-programmed death-1 (anti-PD-1), and anti-PD-ligand-1 (PD-L1) antibodies in different solid tumors has aroused interest to explore the potential therapeutic effect in ACC as well. Multiple ongoing clinical trials are currently evaluating the role of immune checkpoint inhibitors in ACC (pembrolizumab, combination pembrolizumab and relacorilant, nivolumab, combination nivolumab and ipilimumab). The primary and acquired resistance to immunotherapy continue to counter treatment efficacy. Therefore, attempts are made to combine therapy: anti-PD-1 antibody and anti-CTLA-4 antibody, anti-PD-1 antibody and antagonist of the glucocorticoid receptor. The inhibitors of immune checkpoints would benefit patients with antitumor immunity activated by radiotherapy. Immunotherapy is well tolerated by patients; the most frequently observed side effects are mild. The most common adverse effects of immunotherapy are skin and gastrointestinal disorders. The most common endocrinopathy during anti-CTLA treatment is pituitary inflammation and thyroid disorders.

## 1. Introduction

The introduction and widespread use of monoclonal antibodies and tyrosine kinase inhibitors has been the most recent major breakthrough in cancer treatment. The dynamic development of a new research area is the so-called immuno-oncology. Cancer immunotherapy—apart from surgery, radiotherapy, chemotherapy, and molecularly targeted therapies—has been recognized as the basic (and very promising) method of treating malignant neoplasms. Immunotherapy is a treatment strategy based on the activation of a dormant and malfunctioning immune system that does not ensure proper recognition of cancer cells, which leads to the development of neoplastic disease. There have been some impressive contemporary discoveries about immune checkpoints. Cancer cells have been found to have the ability to “trick” the immune system by stimulating immune checkpoints, thereby inhibiting the immune system in the ability to recognize cancer neoantigens or produce enough activated T cells to destroy cancer cells. The discovery of immunocompetent molecules that can inhibit the interaction of a cancer cell with antigen presenting cells or effector cells has resulted in great advances in oncology and oncohematology. Immunotherapeutic agents such as cytokines stimulating the immune system or monoclonal antibodies directed against tumor-cell-surface receptors have been approved. Thorough understanding of the mechanisms by which cancer cells escape from the surveillance of the immune system and ways of stimulating the anti-cancer immune response opened the way to the development of a new generation of drugs—immunotherapeutics, targeting immune checkpoints. These drugs have proven to be particularly effective in the treatment of advanced melanoma, non-small-cell lung cancer, bladder cancer, kidney cancer, and cancers of the head and neck region.

The classic concept of passive immunotherapy refers to the use of monoclonal antibodies directed against the surface receptors of cancer cells. The basic mechanism of action of antibodies is to block receptors for intracellular signaling, thereby inhibiting proliferation and apoptosis of target cells. Antibodies also prevent the homo- or heterodimerization of receptors, which is often necessary for the activation of the intracellular signaling cascade [1]. In addition, the use of IgG1 antibodies (e.g., cetuximab, trastuzumab) allows the destruction of neoplastic cells by antibody-dependent cellular cytotoxicity (ADCC) or by activation of the complement system. Another mode of action is to increase the production of autologous anti-epidermal growth factor receptor (EGFR) antibodies, such as with cetuximab, or anti-HER2 (HER2 human epidermal receptor type 2), such as with trastuzumab.

Antibodies targeting immune checkpoints have a different mechanism of action, consisting of the abolition of immune tolerance. Their toxicity profile and efficacy are also unique—these antibodies have the ability to induce long-term disease control in some patients. The concept of immunotherapy using antibodies against immune checkpoints is their ability to reverse T cell anergy in the tumor environment. Lymphocytes then only receive activation signals and regain their ability to destroy tumor cells. Due to its remarkable effectiveness in the case of some types of cancer, this strategy has revolutionized the existing methods of immunotherapy. This revolution in cancer therapy took place in a very short time and currently, several years after the start of research, monoclonal antibodies such as anti-cytotoxic T cell antigen 4 (anti-CTLA-4), anti-programmed cell death protein 1 (anti-PD-1) and anti-programmed cell death protein ligand 1 (anti-PD-L1) are used in routine treatment or are undergoing the last stages of clinical trials in patients with various types of cancer [2].

## 2. Adrenocortical Carcinoma—Pathophysiology and Treatment

Adrenocortical carcinoma (ACC) is a rare epithelial neoplasm that originates in the adrenal cortex and has a high propensity to local invasion and distant metastases. An annual incidence of ACC is between 0.7 and 2 cases per million population [3]. ACC is more frequent in women (55–60%), with a peak incidence in the fourth and fifth decades of life. Despite much effort to improve care for patients with ACC, diagnostic and therapeutic options remain limited. The most common presenting complaints are symptoms and signs of hormone excess (40–60%) and pain due to local tumor growth (30–40%). Highly differentiated cancer mainly secretes cortisol and androgens in excess, but also produces significant amounts of estrogens and mineralocorticoids. The ability to synthesize biologically active corticosteroids depends on the degree of cellular differentiation. Hormonally active cancer is most often caused by Cushing’s syndrome, with concomitant features of androgenization (often very dynamic in women), which may be overlooked in men. Hypertension and secondary amenorrhea are common in women. Excess estrogen can cause gynecomastia and other feminization symptoms in men, such as a change in the distribution of body fat, and heavy menstrual bleeding in women. Hormonally non-functional cancer appears late, usually with progressive weight loss and symptoms related to the location of distant metastases. Metastases, apart from regional lymph nodes, are most often located in the lungs, liver, and bones [4]. Approximately 10–15% of ACCs are incidentally discovered on imaging studies. 

ACC carries a poor prognosis, with overall five-year survival ranging from 60–80% in patients with ACC stage I, to 13% in patients with stage IV disease [4]. In localized ACC, a complete remission can be achieved, but the manifestation of metastases or recurrence in follow-up significantly worsens the prognosis. Complete remission is extremely rare in patients diagnosed with ACC in the stage of regional invasion or distant metastases. The available treatment regimens for ACC are limited, and radical surgical resection is the only curative option. Unfortunately, most ACC patients present with locally advanced or metastatic disease not amenable to surgical resection. Surgical treatment is the method of choice—removal of the primary tumor and metastases to the highest possible extent [5]. Completeness of resection is one of the most important prognostic factors for disease-free and overall survivals [6,7]. As soon as possible after tumor removal, mitotane is used, which is recommended to most patients, in particular for stage III and IV cancers or with a high Ki-67 proliferation index (>10) [8]. Mitotane is an adrenolytic drug, a synthetic derivative of the insecticide dichlorodiphenyltrichloroethane (DDT). The limitation of treatment with the drug is its toxic effects, including on the bone marrow, liver, skin, gastrointestinal tract, and neuromuscular junction [8]. Moreover, the response to mitotane treatment is observed only in about 30–50% of patients [5]. In unresectable, regionally advanced, or metastatic cancer, apart from mitotane, systemic treatment is also used in specialized oncological centers, e.g., etoposide, doxorubicin, cisplatin (DEP) regimens plus mitotane is considered the standard chemotherapy in advanced ACC [8]. The disease stabilization is achievable in 50 percent of patients treated with chemotherapy; an objective response should be expected only in <25% [9]. Radiotherapy (RT) is mainly used to treat bone metastases and palliating local symptoms. Currently, the adjuvant RT has been shown to benefit patients with ACC. It was demonstrated that that adjuvant RT have significantly improved locoregional control and overall survival [10]. Thus, radiation should be considered as an option in multimodality management of metastatic ACC patients.

The advances in molecular diagnostics have provided multiple prognostic factors and allowed a better understanding of the pathophysiology and molecular changes in ACC [11,12]. The targeted therapy has also been investigated in advanced ACC. Nevertheless, no major therapeutic breakthrough has been made so far. Promising results were obtained only with recombinant human monoclonal antibodies against insulin-like growth factor type 2 receptor (IGF-2R) [13]. IGF-2 is a growth factor secreted mainly by the liver, but also in smaller amounts in the majority of tissues where it acts in an autocrine or paracrine way. IGF-2 actions are mediated by the IGF-1 receptor, insulin receptor, and IGF-2R. IGF-2 activates tyrosine kinase receptors which in turn lead to activation mechanisms involved in the proliferation, survival, and metastasis of cancer cells (mitogen-activated protein kinase (MAPK) and phosphatidylinositol 3-kinase (PI3K)/Akt pathways). At the molecular level, around 90% of adult patients with ACCs exhibit IGF-2 overexpression when compared to normal adrenal glands or adrenocortical adenomas (ACA). IGF-2 mRNA expression was found to be 10–20-fold higher in ACC compared to normal adrenal glands or ACA, while IGF-2 protein expression was described to be 8–80-fold greater in ACC than in normal adrenal glands or ACA. In addition, IGF-2 was demonstrated to influence adrenocortical cancer cell proliferation, metabolism, and viability, but not the cell invasion. However, no significant differences in clinical, biological, and transcriptomic traits were found between tumors with high and low expression of *IGF-2* [13,14]. 

The dysregulation of mammalian targets of the rapamycin kinase (mTOR) pathway and activation of the Wingless and Int-1 (WNT)/β–catenin pathway plays an important role in sporadic adrenocortical tumorigenesis [12,13,15]. The vascular endothelial growth factor receptor (VEGFR) and epidermal growth factor receptor (EGFR) inhibitors are thought to have promising roles in the treatment of ACC [16,17,18]. 

## 3. Immunotherapy in ACC Treatment

Immunotherapy is the latest revolution in cancer therapy; however, preliminary data from studies with single immune checkpoint inhibitors showed a modest activity in ACC patients [9]. The anti-neoplastic activity of immune checkpoint inhibitors such as anti-cytotoxic-T-lymphocyte-associated-antigen 4 (anti-CTLA-4), anti-programmed death-1 (anti-PD-1), and anti-PD-ligand-1 (PD-L1) antibodies in different solid tumors has aroused interest to explore the potential therapeutic effect in ACC as well [19,20,21]. Multiple ongoing clinical trials are currently evaluating the role of immune checkpoint inhibitors in ACC [22] (Table 1).

### 3.1. Immune Checkpoint Inhibitors

IL-13-PE is a recombinant cytotoxin chimeric fusion protein that consists of interleukin -13 (IL-13) and a mutated form of Pseudomonas exotoxin A (PE) [23]. Interleukin-13 receptor alpha2 (IL13Rα2) is a high-affinity receptor of the T-helper 2 cell-derived cytokine IL-13 and is overexpressed in several types of cancers compared with low or absent expression in normal cells and tissues [24]. It has been demonstrated that high-affinity binding of the IL-13 ligand to IL13Ra2 does signal through a STAT6-independent activator protein 1 (AP-1) pathway, which leads to increased transforming growth factor-beta (TGF-b) activity. The studies of Jain et al. have demonstrated the overexpression of (IL13Rα2) in ACC cells [24]. It has been shown that IL-13-PE is highly cytotoxic to IL13Ra2-positive cancer cells in both in vitro and in vivo models of several malignancies. Moreover, it has been demonstrated that IL-13Ra2-positive ACC cell lines were sensitive to IL13-PE as well [25]. In a phase I trial using intravenous infusion of IL-13-PE in ACC patients with IL-13Rα2 expression, stable disease was observed in one-fifth of patients who were treated at maximum-tolerated dose with progression times ranging from 1 to 5.5 months; the others progressed within 1–2 months [25]. All studied patients developed high levels of neutralizing antibodies during IL-13-PE treatment which might have limited the drug efficacy.

Ipilimumab is a recombinant fully humanized monoclonal IgG antibody that is directed against CTLA-4. This is the first drug that gives patients with advanced melanoma a chance for long-term survival [26]. By blocking CTLA-4, the drug stimulates the activation and proliferation of T cells in the lymph nodes [27]. Ipilimumab also reduces the immunosuppressive activity of T cells in the tumor microenvironment by blocking CTLA-4 on regulatory T cells [27]. In addition, this drug increases the percentage of activated helper and cytotoxic T cells in the peripheral blood [28]. There are two ongoing phase II clinical trials evaluating the safety and effectiveness of ipilimumab in combination with nivolumab (anti-PD-1 antibody) in patients with rare genitourinary malignancies including ACC (still recruiting; Table 1). 

PD-1 is an immune-checkpoint receptor expressed by T cells. PD-L1 and PD-L2 are also expressed in the tumor microenvironment of various cancers, including genitourinary tumors. Binding of PD-1 to PD-L1 or PD-L2 negatively regulates T cell effector functions and reduces the immune surveillance of tumor cells [28]. It has been estimated that 11% of ACCs express PD-L1 on tumor cell membranes, and 70% of tumor-infiltrating monocytes are PD-L1-positive [29]. Nivolumab and pembrolizumab are monoclonal antibodies that target the PD-1 receptor on T cells. By blocking PD-1 in the tumor microenvironment, the drugs enable T lymphocytes to recognize tumor cells [29]. This blockade prevents PD-1 from binding to the PD-L1 and PD-L2 ligand on tumor cells, which is the mechanism by which tumor cells escape from immune surveillance. 

Pembrolizumab is a humanized recombinant monoclonal IgG class 4 kappa-isotype antibody to PD-1, and it results in an increased immune reactivity that can overcome immune tolerance, thus enabling its use in immunotherapy [30]. In a phase II study using pembrolizumab in microsatellite-high and/or mismatch repair deficient (MSI-H/MMR-D) ACC patients, the objective response rate to pembrolizumab was 23% (9 patients), and the disease control rate was 52% (16 patients) [31]. Pembrolizumab provided clinically meaningful and durable antitumor activity in ACC patients with tumors that were microsatellite-stable, with a manageable safety profile [31]. In a phase II study of pembrolizumab in patients with advanced rare cancers, 15 patients with ACC were enrolled [29]. The progression-free survival rate at 27 weeks was 31% (four patients); moreover, the objective response rate (ORR) of 15% and clinical benefit rate (CBR) of 54% were clinically significant [29]. In an ongoing phase I study of relacorilant (a small molecule antagonist of the glucocorticoid receptor) in combination with pembrolizumab, eligible patients are those with advanced ACC associated with glucocorticoid excess. The goal of this study is to assess the safety and efficacy of the drugs in this population (Table 1). 

Nivolumab, a fully human immunoglobulin G4 PD-1 immune checkpoint inhibitor antibody, blocks PD-1 and promotes antitumor immunity, and it is effective for treating non-small-cell lung cancer (NSCLC), melanoma, renal cell carcinoma (RCC), and other cancers [32]. In a phase II study, nivolumab demonstrated modest antitumor activity in patients with advanced ACC [33]. The drug was used in 10 patients with metastatic ACC who failed other treatments. Nivolumab therapy was associated with the median time to progression of 1.8 months. The median follow-up was 4.5 months (range, 0.1 to 25.6 months). Two patients had stable disease for 48 and 11 weeks, respectively. The safety profile of nivolumab was consistent with previous clinical experience, without any unexpected adverse events [33].

Avelumab is an investigational fully humanized anti-PD-L1 IgG1 monoclonal antibody [30]. Avelumab is a monoclonal antibody that targets PD-L1. In a phase Ib dose-expansion cohort study, 50 patients with metastatic ACC, who were previously treated with platinum-based therapy, received avelumab every two weeks. Fifty percent of study participants received concomitant therapy with mitotane. A partial response was observed in 3 of the 50 patients (6%), but more importantly 21 patients (42.0%) had stable disease as their best overall response, with a median progression-free survival of 2.6 months and median overall survival of 10.6 months. Avelumab also showed clinical activity and a manageable safety profile [34].

Durvalumab and atezolizumab are monoclonal antibodies directed against PD-L1 [30]. Both drugs are approved for advanced urothelial carcinoma and metastatic non-small-cell lung cancer, but there are currently no clinical trials recruiting ACC patients. 

Due to the mechanism of action of the aforementioned anti-PD-1 and anti-PD-L1 antibodies, the predictor of cancer immunotherapy, determining the effectiveness of treatment, should be PD-L1 expression on the surface of cancer cells and antigen presenting cells [2]. Some studies indicate that positive expression of this protein does not guarantee response to treatment [22,29]. PD-L1 expression was investigated in 28 ACC tissue samples by immunohistochemistry, showing that a small percentage of tumors (10.7%) were PD-L1 expression-positive, with a cut-off level of 5% [35]. Therefore, according to this intrinsic tumor parameter, one would expect the ACC response to immunotherapy to be weak, if anything [36]. It has been shown that immunotherapy can also be effective in patients whose cells do not express PD-L1 [2]. The number of mutations in neoplastic cells also seems to be an important predictor of immunotherapy, targeting immune checkpoints [31]. In cancer patients with tumors characterized by a high number of mutations and neoantigens, as well as high immunogenicity, the response rate is higher, and the response time is longer. Moreover, the majority of ACC patients have a hormone-secreting disease and glucocorticoids are known to exert an immunosuppressive effect [37]. Thus, both endogenous glucocorticoid levels, due to tumor secretion, and glucocorticoid supplementation in patients treated with mitotane have the potential to impair immunotherapy efficacy in ACC patients [38]. Other potential mechanisms of immunoresistance that have been identified are the activation of the WNT-β–catenin pathway or TP53 mutations [36]. Proper identification and the optimal selection of patients who will benefit from anti-cancer immunotherapy remains a challenge and requires further research. The comparison of immune checkpoints is shown in Table 2.

The ongoing research also focuses on the immune checkpoint inhibitors that target novel immune mechanisms. B7-H3 (CD276), part of the B7 superfamily of immune checkpoint molecules, has been shown to have an immunomodulatory role. Its regulation, receptor and mechanism of action remain unclear. B7-H3 protein expression correlates with prostate cancer outcomes, and humanized monoclonal antibodies (enoblituzumab) are currently being investigated for therapeutic use [39]. Lirilumab is a human IgG4 monoclonal antibody that blocks the killer immunoglobulin like receptors (KIR) [40]. Currently, one clinical trial is recruiting patients with thyroid cancer to receive combination lirilumab with nivolumb and ipilimumb (NCT01714739) andipilimumab with enoblituzumab (NCT02381314). There are no ongoing clinical trials using these drugs for patients with ACC.

Due to the promising results of immune checkpoint inhibitor therapies so far, new molecules are constantly being investigated and tested as potential targets of immunotherapy, such as lymphocyte activation gene-3 (LAG-3), T cell immunoglobulin mucin-3 (TIM-3), and indoleamine-2,3- dioxygenase (IDO). Lymphocyte activation gene 3 (LAG-3) is expressed on activated T cells, NK cells, B cells, and tumor-infiltrating lymphocytes, and acts as a negative regulator of T cell activation and homeostasis [41]. LAG-3 blockade resulted in superior T cell activation compared to the inhibition of other pathways, including PD-1/PD-L1 [42]. TIM-3 is a type I transmembrane glycoprotein that is expressed on differentiated T helper type 1 (Th1) but not Th2 cells [43]. TIM-3, after binding with one of its ligands, galectin-9 (Gal-9), leads to cell death, especially in T cells whose activities have been suspended. TIM-3 can also impair immune responses by promoting the expansion of myeloid-derived suppressor cells [44]. Indoleamine 2,3-dioxygenase (IDO) is an enzyme of the immunosuppressive effects that result from its role in tryptophan catabolism [45]. IDO is upregulated in malignancy and is associated with a poor prognosis in multiple cancer types. IDO1 suppresses local CD8+ T effector cells and natural killer cells and induces CD4+ T regulatory cells and myeloid-derived suppressor cells [46]. IDO1 inhibitors have limited activity on their own, but greatly enhance the anti-neoplasm effect of immune checkpoint inhibitors. Clinical trials of IDO inhibitors with chemotherapy or immunotherapy are currently underway [47,48]. 

### 3.2. Combined Therapy

The primary and acquired resistance to this type of treatment, namely, immune checkpoint blockade, continues to counter treatment efficacy. Therefore, attempts are made to combine therapy: anti-PD-1 antibody and anti-CTLA-4 antibody; anti-PD-1 antibody and an antagonist of the glucocorticoid receptor (Table 1). Additionally, the combination of anti-TIM-3 with anti-PD-1 or anti-CTLA-4 shows promise for the improvement of current immunotherapy [42]. It appears that inhibitors of immune checkpoints would benefit patients with antitumor immunity activated by radiotherapy, which can induce an immunogenic cell death and promote the activation of the T cell response [49]. In this proimmunogenic environment, the effectiveness of immunotherapy is significantly higher. Preclinical investigations demonstrated that the radiotherapy followed by pembrolizumab is a well-tolerated treatment with acceptable toxicity [50]. Lenvatinib is an oral, multi-targeted tyrosine kinase inhibitor which impairs several signaling networks implicated in tumor growth and maintenance. It inhibits the vascular endothelial growth factor receptor family (VEGFR1-3), the fibroblast growth factor receptor family (FGFR1-4), platelet-derived growth factor receptor-alpha (PDGFRα), tyrosine-kinase receptor (KIT), and rearranged during transfection receptor (RET) [51]. This drug is indicated for the treatment of locally recurrent or metastatic progressive, radioiodine-refractory differentiated thyroid cancer [52]. It seems that Lenvatinib, via different mechanisms, may synergize the effectiveness of immune checkpoint inhibitors [51]. In particular, preliminary results in patients with selected solid tumors demonstrated that the combination of lenvatinib and pembrolizumab had antitumor activity with partial responses, manageable toxicities, and no new safety signals identified [53]. The combination therapy seems to be a promising alternative in cancer treatment and may have a beneficial effect on the outcomes of patients’ treatment; however, it should be taken into account that this is an active area of investigation. There are currently no open clinical trials because preliminary studies are ongoing, the results of which are being awaited.

### 3.3. Adverse Effects

Overall, immunotherapy is well tolerated by patients; the most frequently observed side effects are mild [31]. Side effects of immunotherapy can occur at any time: in the first weeks after starting treatment, during treatment, as well as after stopping treatment. Although adverse events were common in a study by Kartolo et al. (incidence rate of more than 50%), most patients (70%) developed adverse events of mild-to-moderate severity, and only a small proportion (15%) developed multiple adverse events [54]. Notably, it has been shown that the risk of adverse events depends on factors such as steroid use before immunotherapy, patient sex, a history of autoimmune disease, immunotherapy with a CTLA-4 monoclonal antibody, and kidney dysfunction (poor kidney function (stages 3 and 4 per the U.K. Renal Association) is correlated with a higher risk of side effect development) [54]. Compared to monotherapy, anti-CTLA-4 and anti-PD-1 combination therapy is associated with the highest incidence of immunotherapy-related endocrinopathies (hypothyroidism, hyperthyroidism, hypophysitis, primary adrenal insufficiency) [55]. The management techniques are mainly hormone replacement and symptom control.

The most common adverse effects of immunotherapy are skin disorders—rash, itching, dry skin, erythema, alopecia areata, pruritus, dermatitis, vitiligo, sarcoidosis, and pyoderma gangrenosum (Table 3) [56]. The adverse effects also affect the gastrointestinal tract and include aphthous ulcers, esophagitis, gastritis, and colitis. Serious complications such as bowel perforations are rare but potentially fatal. Severe colitis, complicated by gastrointestinal perforation, was observed during treatment with ipilimumab and pembrolizumab [31]. Other gastrointestinal symptoms include nausea, diarrhea, abdominal pain, peritoneal symptoms, intestinal obstruction, pancreatitis, and hepatitis [33]. 

Immune-related adverse events are frequent complications of immunotherapy, and endocrinopathies are among the most common side effects. The endocrinopathies occur most often during the first weeks of therapy, but the onset of symptoms may be delayed, and hormonal disturbances may appear after the end of immunotherapy [55]. These include hypophysitis, thyroid dysfunction, insulin-deficient diabetes mellitus, and primary adrenal insufficiency. Pituitary inflammation is relatively more common with anti-CTLA-4 agents such as ipilimumab [55,57,58]. The onset of symptoms occurs 6–12 weeks after treatment initiation. Symptoms include a headache (89%), fatigue, muscle weakness, nausea, lack of appetite, and vision disorders [59]. However, these symptoms are nonspecific and could be misattributed to general symptoms related to cancer. In ipilimumab-induced pituitary inflammation, thyrotropic (90%), corticotropic (70%) and gonadotropic (70%) insufficiencies are the most commonly observed disorders; diabetes insipidus has not been observed [60]. Adrenal insufficiency associated with CTLA-4-related hypophysitis is usually permanent, and requires continuous glucocorticoid replacement therapy, while failure of thyrotropic and gonadotropic axes may be transient [59].

Thyroid disorders are more frequently associated with anti-PD-1-antibodies and combination anti-PD-1 and anti-CTLA-4 therapy [55,61]. Immune checkpoint blockers are associated with a high risk of thyroid autoimmunity; this risk is highest for anti-PD-1 and increases further when a combination of checkpoint inhibitors is used [62]. Thyroid dysfunction can present as hypothyroidism, thyrotoxicosis, painless thyroiditis, and transient thyrotoxicosis, followed by hypothyroidism or even “thyroid storm”. Hypothyroidism and thyrotoxicosis related to inflammatory thyroiditis are the most frequent presentations [62]. The severity of the symptoms of thyroid disorders is usually mild hyrotoxicosis which is rarely severe, and hypothyroidism is manageable with thyroid hormone replacement therapy [63]. Thyroid dysfunction in cancer patients may be often unnoticed because it can be asymptomatic or overshadowed by the symptoms of an advanced or progressing cancer. Notably, the thyroid status among cancer patients treated with immune checkpoint inhibitors should be assessed at baseline and periodically after initiation of the immunotherapy. 

Diabetes mellitus is a rare complication of immunotherapy; anti-PD-1 therapy alone or in combination with anti-CTLA-4 increases the incidence of diabetes by 1–3% [64]. The onset of hyperglycemia may occur even after the first infusion. Immunotherapy-induced diabetes is insulin-dependent from the very beginning, and progresses rapidly [55,64]. Primary hypoparathyroidism and primary adrenal insufficiency are less frequent findings on immunotherapy [60]. 

Rheumatic adverse events appear to be less frequent, but are underdiagnosed, less well recognized, and unreportable by many clinical trials. Rheumatic disorders occur both early and late in response to immunotherapy, and a substantial proportion of them are chronic, persisting even after cessation of immune checkpoint inhibitors [65]. The spectrum of rheumatic diseases includes arthralgia, arthritis, enthesitis, myalgia, myositis, sarcoidosis, systemic sclerosis, Sjögren syndrome, lupus, and vasculitis [66]. Arthralgia or myalgia seem to be the most common rheumatic symptoms [67]. Clinical trials of metastatic melanoma showed that the combination of ipilimumab and nivolumab compared with respective monotherapies was associated with higher frequencies of arthralgia and myalgia [68]. Their management consists of anti-inflammatory treatment, including glucocorticoids, synthetic and biologic immunomodulatory/immunosuppressive drugs, and symptomatic therapies. It is rarely required to discontinue the immunotherapy [66].

## 4. Conclusions

There is no doubt that chemotherapy in the treatment of patients with the most advanced cancers has reached or is approaching its limit, especially in ACC patients; new treatment concepts are therefore urgently needed. The discovery of targeting mutations and immune checkpoints led to the rapid development of molecularly targeted therapies and immunotherapy, which could replace chemotherapy in some patients and in some lines of treatment. Cancer immunotherapy, especially targeting immune checkpoints, differs from chemotherapy in that it is less toxic and has the ability to induce long-term disease control in some patients. The response to immunotherapy is not as spectacular as with molecularly targeted methods, but the duration of the response is often much longer. Finally, although we have some evidence regarding the potential benefit of immunotherapy in ACC, there are no data available regarding its use in the adjuvant setting. There is also a need for further research on predictive factors that will allow the groups of patients who can benefit most from the treatment to be precisely defined. Time is also needed—unlike classical cytostatic chemotherapy, cancer immunotherapy is a treatment strategy that does not work immediately after administration. The therapeutic effect may appear too late in patients with advanced disease and metastases. 

## Figures and Tables

**Table 1 biomedicines-09-00098-t001:** Clinical studies investigating immunotherapy in adrenocortical carcinoma (ACC).

Drug	Target	Clinical Trials.Gov Identifier	Study Phase	Status
Pembrolizumab	anti-PD-1 antibody	NCT02673333 NCT02721732	II II	Active, not recruiting Recruiting
Combination pembrolizumab and relacorilant	anti-PD-1 antibody and antagonist of the glucocorticoid receptor	NCT04373265	I	Not yet recruiting
Nivolumab	anti-PD-1 antibody	NCT02720484	II	Terminated
Combination nivolumab and ipilimumab	anti-PD-1 and anti-CTLA-4 antibody	NCT02834013 NCT03333616	II II	Recruiting Recruiting

**Table 2 biomedicines-09-00098-t002:** Comparison of immune checkpoints.

	PD-1	CTLA-4
Expression on cells	Activated B and T cells Activated NK cells (natural killers) Tumor infiltrating lymphocytes (TILs) in various types of cancer	Effector and regulatory T cells activated in the initial phase of antigen response
Function	Inhibitory receptor Reduces the activity of T cells in peripheral tissues after an inflammatory reaction occurs Reduces the autoreactivity of lymphocytes and NK cells	Inhibitory receptor Regulates (reduces) the activation of T cells at an early stage of their differentiation
Ligands	PD-L1 (B7-H1/CD274) PD-L2 (B7-CD/CD273)	CD80 (B7.1) CD86 (B7.2)
Mechanism desciption	The interaction of the receptor with its ligand causes the activation of Src homology region 2 domain-containing phosphatase-2 (SHP-2) and a decrease in the expression of the Bcl-xL protein, which results in the inhibition of the activity of PI3K/AKT kinase	The interaction of the receptor with its ligand causes the activation of SHP-2 and proteine phosphatase 2 (PPA2) and the blockade of the expression of various proteins in the cell membrane and the flow of Ca^2+^ ions, which results in the inhibition of signal transmission by T-cell receptor (TCR) (blockade of ZAP70 protein formation)

**Table 3 biomedicines-09-00098-t003:** Adverse events with immune checkpoint inhibitors.

	Adverse Event
Cutaneus	rash, itching, dry skin, erythema, alopecia areata, pruritus, dermatitis, vitiligo, sarcoidosis, pyoderma gangrenosum
Gastrointe-stinal	nausea, diarrhea, abdominal pain, peritoneal symptoms, intestinal obstruction, pancreatitis, hepatitis, aphthous ulcers, esophagitis, gastritis, colitis, oral mucositis
Endocrine	thyroid dysfunction (hyperthyroidism, hypothyroidism), adrenal insufficiency, hypophysitis, hypopituitarism, diabetes mellitus, primary hipoparathyroidism
Rheumato-logic	arthralgia, arthritis, enthesitis, lupus, sarcoidosis, systemic sclerosis, Sjögren’s syndrome, vasculitis, myalgia, myositis
Pulmonary	pneumonitis, sarcoidosis
Renal	nephritis (interstitial, glomerulonephritis)
Neurologic	peripheral neuropathy, headache, dizziness, Guillan–Barre syndrome, polyneuropathy, autoimmune encephalitis, myasthenia gravis

## Data Availability

Not applicable.

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
