# Peer review of "The Role of Immunotherapy in the Treatment of Adrenocortical Carcinoma"

_biomedicines, 2021, doi:10.3390/biomedicines9020098_

Round 1

Reviewer 1 Report

It is very informative to summarize immunotherapy in the treatment of adrenocortical carcinoma. The reviewer have several comments and suggestions:

1. Introduction: It is well written. 

2. Adrenocortical carcinoma - pathophysiology and treatment: It is well written as well. I would like to know about IFG-2R more. Please write more detail on IGF-2R including gene function.

3-1 Immunotherapy in ACC treatment: This section must be the best important in this article. A figure describing this section is better to have. Especially, many readers do not know IL-13-PE. Please describe the molecule more descriptively.

3-2 better to have a figure as well.

3-3 This portion seems to be describing general things on immunotherapy but not on immunotherapy for adrenocortical carcinoma. It must be short or not needed.

Author Response

Thank you for taking the time to review our article. Your comments were included in the corrected version of the article.

Kind regards,

Izabela Karwacka

Reviewer 2 Report

Summary
Karwacka and colleagues summarize the current evidence, ongoing trials, and adverse reactions to immunotherapy and in the treatment of adrenocortical carcinoma. The article is well-written and easy to follow. Minor comments below.

2. Adrenocortical carcinoma - pathophysiology and treatment, page 2, line 93 and page 2, line 95, please change “recovery” to “remission”

2. Adrenocortical carcinoma - pathophysiology and treatment, page 3, line 107, please add “unresectable” to “regionally advanced or metastatic” since EDP plus mitotane is indicated for unresectable advanced disease.

2. Adrenocortical carcinoma - pathophysiology and treatment, page 3, line 109, “Current chemotherapy…” is verbatim from the abstract of reference 9. Either the authors should place quotes around the sentence to indicate that this is verbatim or paraphrase.

2. Adrenocortical carcinoma - pathophysiology and treatment, page 3, line 111, although still in the research phase, given the rarity of adrenocortical carcinoma, the authors should cite that adjuvant radiotherapy has been shown to have a benefit in patients with adrenocortical carcinoma (PMID: 31220287).

3.2 Combine therapy, page 6, line 241, this entire paragraph appears to be speculative about the possibility of using immune checkpoint inhibitors in combination for patients with adrenocortical carcinoma. The authors should cite and discuss evidence that this strategy is relevant to patients with adrenocortical carcinoma or state that this is an active area of investigation.

3.2 Combine therapy, page 6, line 260, “arktoand” appears to be a typo.

Author Response

Thank you for your time to review our article. Your comments were taken into account in the revised version of the article.

Best regards,

Izabela Karwacka